# Role of Glucocorticoid Signaling and HDAC4 Activation in Diaphragm and Gastrocnemius Proteolytic Activity in Septic Rats

**DOI:** 10.3390/ijms23073641

**Published:** 2022-03-26

**Authors:** Álvaro Moreno-Rupérez, Teresa Priego, María Ángeles González-Nicolás, Asunción López-Calderón, Alberto Lázaro, Ana Isabel Martín

**Affiliations:** 1Department of Physiology, School of Medicine, Complutense University of Madrid, 28040 Madrid, Spain; alvmor02@ucm.es (Á.M.-R.); ALC@med.ucm.es (A.L.-C.); alberlaz@ucm.es (A.L.); 2Department of Physiology, Faculty of Nursing, Physiotherapy and Podiatry, Complutense University of Madrid, 28040 Madrid, Spain; tpriegoc@med.ucm.es; 3Renal Physiopathology Laboratory, Department of Nephrology, Instituto de Investigación Sanitaria Gregorio Marañón, Hospital General Universitario Gregorio Marañón, 28007 Madrid, Spain; mangeleg@ucm.es

**Keywords:** sepsis, inflammation, muscle wasting, glucocorticoids signaling, HDAC4-myogenin, IGF-1, IGFBP3, atrogens, autophagy

## Abstract

Sepsis increases glucocorticoid and decreases IGF-1, leading to skeletal muscle wasting and cachexia. Muscle atrophy mainly takes place in locomotor muscles rather than in respiratory ones. Our study aimed to elucidate the mechanism responsible for this difference in muscle proteolysis, focusing on local inflammation and IGF-1 as well as on their glucocorticoid response and HDAC4-myogenin activation. Sepsis was induced in adult male rats by lipopolysaccharide (LPS) injection (10 mg/kg), and 24 h afterwards, rats were euthanized. LPS increased TNFα and IL-10 expression in both muscles studied, the diaphragm and gastrocnemius, whereas IL-6 and SOCS3 mRNA increased only in diaphragm. In comparison with gastrocnemius, diaphragm showed a lower increase in proteolytic marker expression (atrogin-1 and LC3b) and in LC3b protein lipidation after LPS administration. LPS increased the expression of glucocorticoid induced factors, KLF15 and REDD1, and decreased that of IGF-1 in gastrocnemius but not in the diaphragm. In addition, an increase in HDAC4 and myogenin expression was induced by LPS in gastrocnemius, but not in the diaphragm. In conclusion, the lower activation of both glucocorticoid signaling and HDAC4-myogenin pathways by sepsis can be one of the causes of lower sepsis-induced proteolysis in the diaphragm compared to gastrocnemius.

## 1. Introduction

Sepsis is a systemic inflammatory disease induced by serious infection, which leads to vascular, metabolic, and endocrine disorders, tissue damage, and multiorgan failure. This systemic inflammation can be caused by different microorganisms, generally bacteria, and/or their endotoxins [1]. Similar to other chronic diseases, such as cancer, chronic kidney disease, and diabetes, sepsis induces skeletal muscle atrophy and wasting [2]. Normal muscle protein synthesis rate, together with a dramatic increase in protein degradation, has been observed in critically ill patients with sepsis, as well as in mice after cecal ligation and puncture [3,4]. A decreased protein synthesis rate has also been reported in experimental models of sepsis [5,6,7]. Sepsis-induced muscle atrophy in septic patients is a very serious problem since it increases mortality and morbidity, as well as chronic muscle weakness and functional limitations [8]. In fact, the vast majority of sepsis patients showed mobility problems one year after discharge [9]. These problems highlight the importance of understanding the mechanisms involved in muscle atrophy during sepsis.

It has been postulated that sepsis rapidly produces myopathy, affecting both respiratory and limb muscles [10,11]. However, respiratory and peripheral muscles respond differently to sepsis, where locomotor muscles are more affected than respiratory muscles. In this sense, respiratory muscle weakness was less marked than peripheral muscle weakness, even if the patients received controlled mechanical ventilation [10]. Twenty-four hours after sepsis induced by cecal ligation and puncture, there was a transient decrease in cross-sectional area fibers of the diaphragm, but not in its mass [4,7]. In contrast, a progressive decrease in both muscle mass and the fiber cross-sectional area during seven days after sepsis induction has been observed in limb muscles [4,6,7]. There are several factors that can mediate sepsis-induced muscle wasting, muscle inflammation, increased autophagy, upregulation of the ubiquitin-proteasome system, and increased calpain activity and mitochondria dysfunction [6,7,12,13]. All these factors are induced by sepsis in both the diaphragm and limb muscles, but these responses in the diaphragm are smaller and transient in comparison with those of limb muscles, which are stronger and prolonged [6,7].

There are multifactorial mechanisms by which sepsis induces muscle wasting. Inflammation induced by sepsis increases both systemic and muscle proinflammatory cytokine expression, which are well-known inductors of catabolism and muscle wasting [14]. Therefore, low induction of inflammatory cytokines in respiratory muscles can be the cause of low wasting in these muscles. In addition to the direct effects of cytokines on muscle cells, sepsis, as well other systemic diseases, triggers modifications in the neuroendocrine system, which in turn also modulate skeletal muscle physiology [15]. Endocrine response to inflammation is characterized by a downregulation of insulin and insulin-like growth factor-1 (IGF-1), the primary anabolic signaling targeted in the skeletal muscle [16,17,18]. In addition to the down regulation of anabolic hormones, sepsis increases the secretion of glucocorticoids, which are the main catabolic hormones in skeletal muscle.

Lipopolysaccharide (LPS) injection is one of the most used experimental models of sepsis. LPS, the major component of gram-negative bacterial membrane, induces almost all the pathological consequences of sepsis. LPS activates gastrocnemius proteolysis, decreased serum and muscle IGF-1 levels and stimulates hypothalamic CRH, pituitary ACTH, and circulating corticosterone in rats [19]. In the present study, we analyze the effect of sepsis induced by LPS on the expression of muscle-specific E3 ligases, atrogin-1, and MuRF1 as activity markers of the ubiquitin-proteasome system in the diaphragm and in gastrocnemius muscles. Microtubule-associated protein light chain 3b (LC3b) is an important protein involved in autophagy [20]. When autophagy is induced, cytosolic LC3b (LC3b-I) is conjugated and forms lipidated LC3b (LC3b-II). LC3b-II binds to the expanding isolation membrane and remains bound to complete autophagosome. Therefore, the LC3b-II/LC3b-I ratio was also measured as an index of autophagy activity in both skeletal muscles.

Differences in the endotoxin response between gastrocnemius and diaphragm muscles cannot be directly dependent on systemic IGF-1 or glucocorticoids changes in plasma but can be the result of interactions and signaling across different local biocompartments. Although glucocorticoids are one of the circulating factors with powerful catabolic activity in skeletal muscle [21], it has been reported that muscle response to glucocorticoids differs by muscle type [22]. Accordingly, the glucocorticoid receptor (GR), Kruppel-Like Factor 15 (KLF-15) and regulated in development and DNA damage responses 1 (REDD1) expression were measured as intracellular mediators of glucocorticoid effects on muscle atrophy. Local IGF-1 and its binding proteins (IGFBPs), as well as the IGF-1 receptor (IGF-1R) were also studied in the gastrocnemius and diaphragm.

Several findings have shown that histone deacetylases (HDAC) play a key role in muscle atrophy induced by both glucocorticoids and aging [21,23]. An increase in HDAC4 expression has been found in the skeletal muscle of aged rats, whereas different treatments that block aging-induced HDAC4 up-regulation are able to prevent muscle atrophy [24,25]. The effects of HDAC4 on muscle wasting seem to be mediated, at least in part, by myogenin activation [26]. In fact, activation of the HDAC4-myogenin axis has been reported in several models of muscle atrophy [26,27]. However, it has not been studied so far in any model of LPS-induced muscle atrophy. For that reason, the effect of LPS on the HDAC4-myogenin axis was analyzed in order to elucidate its possible role in the different LPS responses between the gastrocnemius and diaphragm.

Therefore, the aim of our study was to elucidate the mechanisms involved in the different sepsis-induced atrophy between diaphragm and gastrocnemius muscles. We have focused on local inflammation and IGF-1-related mechanisms as well as on the response of these muscles to glucocorticoids and their HDAC4-myogenin axis activation.

## 2. Results

As shown in Table 1, lipopolysaccharide (LPS) administration decreased body weight gain (*p* < 0.01) and tended to increase diaphragm weight, whereas gastrocnemius mass was not significantly modified 24 h after LPS injection. Serum concentrations of creatinine, urea, and lactate were increased in the rats injected with LPS (*p* < 0.01, Table 1). LPS induced a significant decrease in serum levels of glucose and bicarbonate (*p* < 0.01), whereas arterial PaCO_2_, PaO_2_, SO_2_, and pH were not significantly modified by endotoxin (Table 1).

The mRNA levels of E3 ubiquitin ligases, atrogin-1 and MuRF1, and the autophagy marker LC3b (Figure 1A–C) were increased in the diaphragm (*p* < 0.05) as well as in the gastrocnemius (*p* < 0.01) of the rats treated with LPS. A higher stimulatory effect of LPS on atrogin-1 (*p* < 0.01) and LC3b (*p* < 0.05) mRNA levels was observed in gastrocnemius in comparison with the diaphragm, whereas the increase in MuRF1 mRNA levels was similar in both muscles. Autophagy activation was also measured as the LC3b II/I ratio (Figure 1D). This ratio increased significantly (*p* < 0.05) in the gastrocnemius of the animals treated with LPS, but not in their diaphragms, where no changes were detected. Ubiquitinated protein levels were also measured by Western blot but were not modified by LPS injection neither in gastrocnemius nor in diaphragm (data not shown).

Figure 2 shows that LPS upregulated mRNA levels of TNFα and IL-10 in both diaphragm and gastrocnemius muscles, but the LPS-induced increase was much higher in the case of IL-10 (Figure 2A,B). LPS also increased IL-6 mRNA levels in the diaphragm (*p* < 0.5 Figure 2C). In gastrocnemius muscle, this cytokine showed a tendency to increase in rats 24 h after LPS injection. However, these changes did not reach statistical significance (*p* = 0.06). Similarly, LPS also increased the expression of the suppressor of cytokine signaling 3 (SOCS3) in the diaphragm (*p* < 0.05, Figure 2D), whereas SOCS3 mRNA levels were not significantly modified in gastrocnemius muscle.

The effect of LPS administration on glucocorticoid signaling is shown in Figure 3. LPS administration did not modify GR mRNA levels in the diaphragm or gastrocnemius (Figure 3A). However, an increase in both KLF15 (*p* < 0.01, Figure 3B) and REDD1 (*p* < 0.05, Figure 3C) mRNA levels was observed in the gastrocnemius of the rats injected with LPS. In contrast, KLF15 and REDD1 mRNA levels in the diaphragm were similar in the rats injected with LPS and in the control rats injected with saline. In rats injected with LPS, the expression levels of these two factors were significantly higher in gastrocnemius than in their diaphragms.

Figure 4 shows the effect of LPS on the IGF-1 system in the diaphragm and gastrocnemius. LPS administration increased IGF-1R expression in both diaphragm and gastrocnemius (*p* < 0.05, Figure 4A). However, the effect of LPS on IGF-1 mRNA expression in muscle was different in these two muscles, since a decrease in IGF-1 mRNA expression was observed in the gastrocnemius (*p* < 0.01, Figure 4B,D), but not in the diaphragm of the rats treated with LPS. In fact, the expression levels of IGF-1 were significantly lower in the gastrocnemius of LPS-injected rats in comparison with the levels measured in their diaphragms. LPS injection decreased IGFBP-3 mRNA levels in the diaphragm (*p* < 0.05, Figure 4C), whereas LPS did not modify this IGF-1 binding protein in the gastrocnemius.

HDAC4 and myogenin expression in the diaphragm and gastrocnemius of the rats treated with LPS or saline are shown in Figure 5. HDAC4 mRNA and protein levels were increased in the gastrocnemius of the rats injected with LPS (*p* < 0.01, Figure 5A,C), whereas the expression of this enzyme was not significantly modified 24 h after LPS injection in the diaphragm. In the animals injected with LPS, HDAC4 mRNA was higher in the gastrocnemius compared to the expression in the diaphragm (*p* < 0.05, Figure 5A). In contrast, myogenin mRNA levels increased both in the diaphragm and gastrocnemius of the rats that received LPS (Figure 5B), whereas myogenin protein levels were increased only in the gastrocnemius (*p* < 0.01, Figure 5D). These differences may be due to post-transcriptional changes.

## 3. Discussion

In our data, 24 h after LPS administration, a decrease in body weight, hypoglycemia, and hyperlactatemia were observed. Although hyperlactatemia is usually associated with hypoxemia, it does not seem to be the case in this study, since both PO_2_ and SO_2_ levels in blood were similar in LPS and in controls rats. Similar results have been previously reported both in experimental sepsis and in septic patients [28,29]. These and other data suggest that lactate is an aerobically produced metabolite that has been released as a consequence of accelerated aerobic glycolysis and the stress response induced by sepsis (for review see [30]). The increase in serum lactate can also be due to the amino acid conversion to pyruvate as a result of the increase in muscle proteolysis. The fact that the LPS treated rats had higher urea and creatinine levels in serum than control rats supports the hypothesis that hyperlactatemia and hyperuremia can be secondary to increased muscle proteolysis. Similarly, hypoglycemia, hyperlactatemia, and hyperuremia, together with muscle wasting, have been reported in muscle wasting induced by cancer in rats [31].

Increased muscle proteolysis by the ubiquitin-proteasome system was evident 24 h after LPS administration in gastrocnemius muscle, since endotoxins increased the mRNA of two proteolytic markers, the E3 ubiquitin ligases MURF1 and atrogin-1. In addition, LPS increased the expression of the autophagy marker LC3b mRNA, as well as the lipidation of the LC3b protein, measured as the LC3bII/LC3bI ratio. All these data indicate an activation of the ubiquitin-proteasome system as well as of autophagy in the gastrocnemius. In the diaphragm, LPS also increased the expression of the MURF1, atrogin-1, and LC3b mRNA, but the increases in atrogin-1 and LC3b mRNA expression were lower than those observed in the gastrocnemius. Furthermore, in the diaphragm, no increase in the LC3b lipidation was detected 24 h after LPS injection, suggesting no activation of the autophagy. In agreement with previous data reported by other authors [4,6,32], our results suggest that sepsis induces muscle autophagy and proteolysis preferentially in locomotor muscles compared with respiratory muscles, such as the diaphragm. The different effect of sepsis on these two muscles not only affects these processes, but also affects them in different manners, time-dependent and muscle-specific, than the myofribrillar protein expression [7].

One possible factor that could explain the lower proteolysis in the diaphragm than in the gastrocnemius could be the local inflammatory responses in the two muscles. LPS injection increased the expression of the proinflammatory cytokine TNFα in both muscles, as well as that of the anti-inflammatory cytokine IL-10. Unlike TNFα and IL-10, IL-6 mRNA expression was increased in the diaphragm but not in the gastrocnemius of the rats injected with LPS. An increase in IL-6 expression in skeletal muscle has been observed in several models of muscle atrophy [33,34]. However, upregulation of IL-6 does not necessarily induce muscle wasting, since plasma concentration of IL-6 rises during physical exercise and is associated with an increase in muscle mass [35]. Furthermore, acute liver injury does not activate atrophic pathways in a mouse’s diaphragm, although it upregulates IL-6 mRNA [36]. As these authors suggest, the increased IL-6 mRNA levels in the diaphragm could be triggered by the overload of this muscle by the increased work of breathing due to metabolic disturbances after liver injury or sepsis. There are also more data suggesting that IL-6 has little effect on muscle wasting [37]. In this sense, acute IL-6 administration induces protein synthesis in myotube cultures [38]. Another mechanism by which IL-6 can increase muscle mass can be by decreasing the inhibitory effect of glucocorticoid, since the role of IL-6 in experimental asthma-induced glucocorticoid insensitivity has been recently reported [39].

SOCS3 belongs to a family of proteins characterized by the ability to negatively modulate cytokine signaling. SOCS3 is an important factor that regulates inflammation by inhibiting the response of other stimuli, such as LPS and NF-kB signaling pathways [40,41]. Therefore, the higher expression of SOCS3 in the diaphragm after LPS injection may decrease the stimulatory effect of TNFα on atrogenes and autophagy in this muscle.

Other factors that mediate muscle wasting in sepsis are hormones whose secretion is modified by sepsis, such as glucocorticoids and IGF-1 [15]. Differences in the endotoxin response between gastrocnemius and diaphragm muscles are not directly dependent on systemic IGF-1 or glucocorticoids, but rather involved in the interactions and signaling across different biocompartments. Glucocorticoids and glucocorticoid signaling in skeletal muscle are essential for endotoxin-induced cachexia [42]. The fact that glucocorticoid induces atrophy preferentially in fibers that have high glycolytic and low oxidative capacity supports the predominant involvement of limb muscles over respiratory muscles [43]. In our data, GR expression was not modified 24 h after LPS administration. However, the expression of the KLF15 and REDD1 was increased in the gastrocnemius but not in diaphragm, which suggests that the diaphragm is less sensitive to glucocorticoids than the gastrocnemius. In skeletal muscle, GR activation induces the upregulation of KLF15, leading to an increase in atrogin-1 and MuRF1 expressions and activation of the ubiquitin proteasome system [44]. Furthermore, in the liver, KLF15 upregulates the expression of enzymes involved in amino acid catabolism and increases gluconeogenesis, leading to an increase in glucose release by the liver [45]. REDD1 is a ubiquitous and conserved protein that can be found in several cell compartments included in the mitochondria-associated ER membranes [46]. This protein is another glucocorticoid target that mediates its catabolic effect on skeletal muscle [44,47], and whose deletion prevents dexamethasone-induced skeletal muscle atrophy [48]. In contrast to KLF15, REDD1 does not seem to be involved in glucocorticoid-induced atrogin-1 and MuRF1 expression, but rather in autophagy activation [46].

IGF-1R mRNA expression was increased in both gastrocnemius and diaphragm muscles. An increase in the expression of this receptor has been previously reported in other experimental conditions of muscle wasting, such as chronic heart failure, cancer, and adjuvant-induced arthritis [49,50,51]. The similar response of IGF-1R in the two muscles suggests that the different activation of the proteolytic systems is not secondary to their response to systemic IGF-1. In contrast to IGF-1R, IGF-1 mRNA levels were decreased by LPS in the gastrocnemius, but not in the diaphragm. The inhibitory effect of sepsis on IGF-1 expression in locomotor muscles is well known [15,52]. Glucocorticoids are able to decrease muscle IGF-1 both in vivo and in vitro [17,21,53]. The lack of decrease in IGF-1 mRNA in the diaphragm of the rats injected with LPS can be the result of the lower sensitivity of this muscle to the action of glucocorticoids. Local IGF-1 is an important regulator of muscle mass by stimulating protein synthesis and inhibiting proteolysis. Therefore, the decrease in muscle IGF-1 can be another cause of the higher proteolysis in the gastrocnemius than in the diaphragm.

In addition to the liver, IGFBP-3 is expressed in other organs, including skeletal muscle, but its role in cellular physiology is still not well known. In contrast with present data, in which LPS did not modify IGFBP-3 in the gastrocnemius, we have previously observed that LPS upregulated IGFBP-3 in the gastrocnemius [17,19]. Differences in those data can be due to the different experimental protocols, since LPS dosage (10 vs. 0.25 mg/kg) and the time after LPS injection (24 vs. 4 h) were different. Nevertheless, in the present data, the effect of LPS on IGFBP-3 expression was also different in the gastrocnemius than in the diaphragm, since LPS decreased IGFBP-3 expression in the diaphragm, but not in the gastrocnemius. Several data suggest that IGFBP-3 plays an important role in human pancreatic cancer-induced cachexia [54,55]. An increase in muscle IGFBP-3 has also been observed in cachexia induced by experimental models of arthritis and cancer [56,57]. One of the mechanisms by which IGFBP-3 induces muscle wasting seems to be due to an increase in ubiquitin-proteasome proteolysis activity [54]. The fact that LPS decreased IGFBP-3 expression in the diaphragm can contribute, together with the other differences, to lower proteolysis in this muscle than in the gastrocnemius.

Histone deacetylases (HDACs) are known to play a key role in limb muscles atrophy induced by aging [23]. In the diaphragm, an increase in HDC4 levels of patients with chronic obstructive pulmonary disease, together with dysfunction of this respiratory muscle, has been reported [58]. To our knowledge, there are no data on the effect of sepsis on HDAC4 expression in skeletal muscle. However, LPS upregulates HDAC4 in mice podocyte cells MPC5 [59], and these authors concluded that HDAC4 mediates LPS-induced acute renal injury and podocyte injury. Furthermore, TMP195, an HDAC4 inhibitor, attenuates endotoxin-induced acute renal injury [60]. A relationship between HDAC4 and increased activity of glucocorticoid signaling has been reported in stress-induced hyperalgesia, where GR was reversibly acetylated by HDAC4 [61]. Those authors proposed that HDAC4 is recruited by GR activation. In this sense, HDAC4 expression has been reported to be increased by glucocorticoid in the osteoblast [62,63]. Therefore, the lower LPS-induced increase in HDAC4 expression in the diaphragm rather than in the gastrocnemius can be secondary to the lower glucocorticoid sensitivity of this muscle. Similarly, to HDAC4, LPS increased myogenin levels in the gastrocnemius, but not in the diaphragm. Accordingly, in other models of muscle atrophy, an increase of HDAC4 and myogenin has been reported [64]. Furthermore, some authors describe a key role of the overexpression of these two proteins in the induction of muscle atrophy [65]. Activation of the HDAC4 myogenin pathway contributes to further inducing E3 ubiquitin ligases MuRF1 and atrogin-1 [65]. The fact that diaphragm HDAC4 expression was not increased by LPS as it was in gastrocnemius, can be another factor that contributes to lower muscle wasting in respiratory muscles than in locomotor muscles.

In conclusion, the diaphragm of septic rats shows lower expression of proteolytic system markers (atrogin-1, LC3b) and IGFBP-3, and higher levels of myogenic factor (local IGF-1) in comparison with the gastrocnemius. These differences can be due to higher IL-6 activation and lower activation of both the glucocorticoid signaling pathway and HDC4-myogenin axis in this muscle. These two final mechanisms seem to be crucial to explain the different LPS-induced atrophic response of the diaphragm in comparison to locomotor muscles, and they seem to be connected. To our knowledge, this is the first time that it is described that the HDAC4-myogenin axis is activated during sepsis in skeletal muscle.

## 4. Materials and Methods

### 4.1. Animals and Experimental Protocol

Studies were performed on male Wistar rats (Charles River, Wilmington, MA, USA) weighing 150–200 g. They were housed under a temperature and humidity-controlled environment with a 12 h light-dark cycle and free access to food and water. The animals were weighed at the beginning of the procedure and just before the end. All the procedures were approved by the Animal Experimentation Ethics Committee of the Gregorio Marañón Hospital (protocol code PROEX 089/18) and performed in accordance with Directive 2010/63/EU and Spanish Royal Decree 118/2021 on the protection of animals used for experimentation and other scientific purposes.

The study was based on 19 animals randomly distributed into 2 groups: untreated controls (n = 9) and LPS-injected rats (n = 10). LPS 10 mg/kg (from Escherichia coli O55:B5, Sigma Aldrich, St Louis, MO, USA), or its vehicle (0. 9% saline serum) was administered to the rats intraperitoneally (i.p.) in the same manner and volume (5 mL/kg). This high LPS dosage was chosen since it has been reported to induce hypoglycemia, hyperlacticacidemia, and other deteriorations typical of endotoxic shock [66]. The survival rate was 80% 24 h after LPS injection, and 100% in the rats injected with saline. Twenty-four hours after LPS or saline injection, the animals were anesthetized with ketamine (10 mg/kg) and diazepam (4 mg/kg) and euthanized by exsanguination by abdominal aortic puncture. Arterial blood gas parameters were measured immediately in the blood samples on a GEM Premier 3000 (Instrumentation Laboratory, Werfen, Barcelona, Spain). Urea and creatinine were determined in serum and urine using a modular AutoAnalyzer Cobas 711 (Roche, Basel, Switzerland). The creatinine clearance rate was used for GFR determination. Diaphragms and right gastrocnemius were collected, dissected, snap-frozen in liquid nitrogen, and kept at –80 °C until analyses were performed.

### 4.2. Quantitative Real-Time Polymerase Chain Reaction (RT-qPCR)

RNA from diaphragm and gastrocnemius muscles was extracted using the Trisure (BIOLINE, London, UK) method under the manufacturer’s conditions. Quantification and 260/280 ratio, used as a reference for RNA purity, were determined by a BioPhotometer spectrophotometer (Eppendorf International, Hamburg, Germany). RNA integrity was checked on an agarose gel electrophoresis stained with GelRed (Biotium, Hayward, CA, USA).

One microgram of total RNA was used to generate first strand complementary DNA (cDNA) with a High-Capacity cDNA Reverse Transcription Kit (Applied Biosystems, Thermo Fisher Scientific, Madrid, Spain). Diluted cDNA from this reaction, forward and reverse specific primers (Table 2) (300 nM, from Roche Diagnostics, Madrid, Spain), and 1 × Takara SYBR Green Premix Ex Taq (Takara BIO Inc., Otsu, Japan) were used to perform real-time PCR. PCR cycles were: 95 °C for 10 min, 95 °C for 15 s, and 60 °C for 1 min (total 40 cycles). A melting curve was also performed in order to verify the specificity of the amplification. The relative expression was calculated using the 2^−∆∆Ct^ method; *18S* RNA was used as the housekeeping gene.

### 4.3. Protein Analysis by Western Blot

Gastrocnemius and diaphragm samples were homogenized in lysis buffer based on radioimmunoprecipitation assay (RIPA) buffer 10 µL/mg, supplemented with a protease inhibitor mix (phenylmethane sulfonyl fluoride 100 mM, sodium deoxycholate 12.5 mM, sodium orthovanadate 12.5 mM, and with phosphatase inhibitors, all from Sigma-Aldrich, St. Louis, MO, USA). Lysates were incubated for 20 min at 4 °C and centrifugated at 13,000 r.p.m. for 20 min. Protein concentration was estimated using the Bradford protein assay (Sigma-Aldrich, St. Louis, MO, USA).

A mix of protein extracted and Laemmli loading buffer (Bio-Rad, Madrid, Spain) (1:1) was boiled at 95 °C for 5 min. Equal amounts of protein (50 µg) were electrophoretically separated on polyacrylamide 4%–20% gradient gels (Bio-Rad, Madrid, Spain) under reducing conditions at 100–200 V for 90 min. Separated proteins were transferred onto a nitrocellulose membrane and then blocked using 5% non-fat dry milk and 0.1% Tween (Sigma-Aldrich, Madrid, Spain) in Tris-buffered saline. In order to analyze the transfer efficiency, membranes were stained with Ponceau-S (Bio-Rad, Madrid, Spain). After blocking the membranes with Tris-buffered saline (TBS) containing 5% (*w/v*) non-fat dried milk, they were incubated overnight at 4 °C with primary antibodies: HDAC4 (antibody ID: 7628, 1:2000; Cell Signaling Technology; Danvers, MA, USA); Myogenin (antibody ID: sc-12732, 1:500; Santa Cruz Biotechnology; Dallas, TX, USA); LC3b (antibody ID: 12741, 1:1000; Cell Signaling Technology; Danvers, MA, USA). Secondary antibodies conjugated to horseradish peroxidase used to detect primary antibodies were: anti-mouse immunoglobulin G (IgG) (Amersham Biosciences; Little Chalfont, UK); anti-rabbit IgG (GE Healthcare; Chicago, IL, USA). Bands were visualized using peroxidase activity (enhanced chemiluminescent reagent from Amersham Biosciences, Little Chalfont, UK). Band intensities were analyzed by densitometry using Gene Tools Analysis software.

### 4.4. Statistical Analysis

Values are expressed as means ± standard error of the mean (SEM), and differences among groups were analyzed by Student’s t-test. All analyses were performed using SPSS 25 for Windows. A *p* value of <0.05 was considered significant.

## Figures and Tables

**Figure 1 ijms-23-03641-f001:**
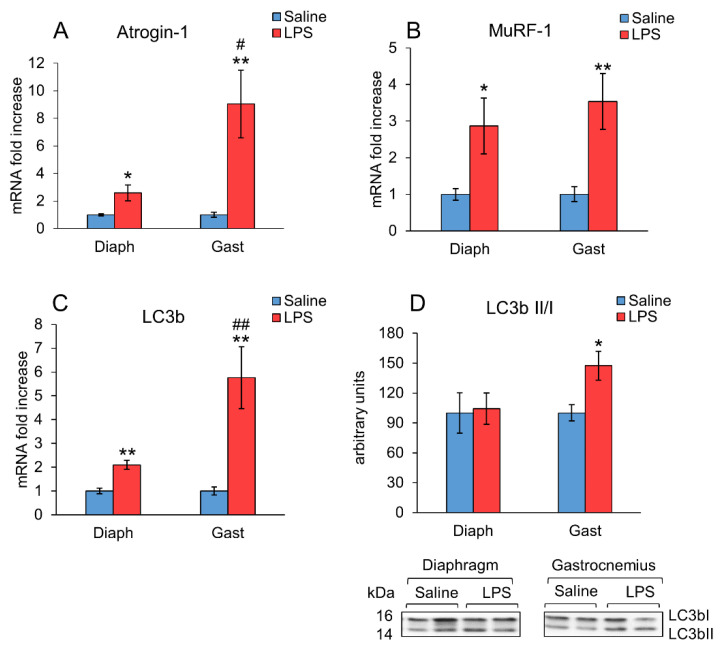
Effect of sepsis induced by lipopolysaccharide (LPS) injection (10 mg/kg ip.) on (**A**) Atrogin-1 mRNA, (**B**) MuRF1 mRNA, (**C**) LC3b mRNA, and (**D**) LC3b-I, and its lipidated form LC3b-II, in rat diaphragm and gastrocnemius muscles, 24 h after a single injection of LPS or saline solution. mRNA was measured by PCR and proteins by Western blot. Representative Western blots are shown in D (bottom). Data represent mean ± standard error of the mean (SEM) for n = 7–8 rats/group. * *p* < 0.05 and ** *p* < 0.01 versus their respective control rats treated with saline, # *p* < 0.05 and ## *p* < 0.01 versus the diaphragm of rats treated with LPS. Diaph: diaphragm, Gast: gastrocnemius.

**Figure 2 ijms-23-03641-f002:**
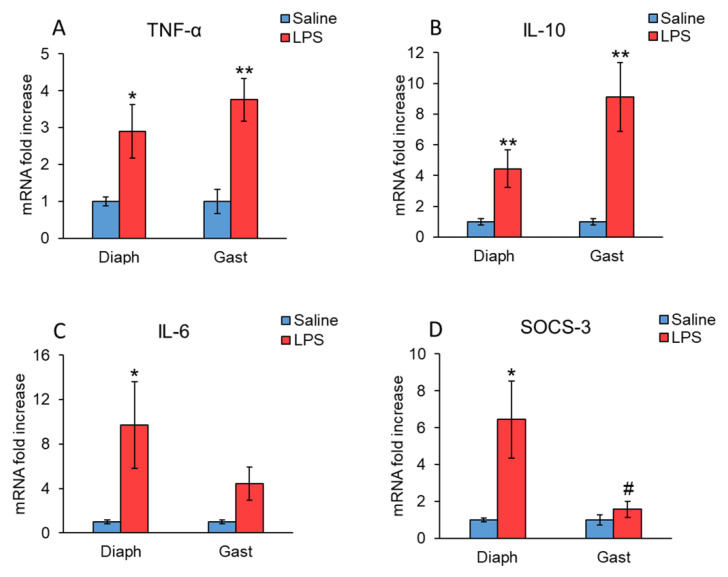
Effect of sepsis induced by LPS injection (10 mg/kg ip.) on (**A**) TNFα, (**B**) IL-10, (**C**) IL-6 and (**D**) SOCS3 mRNA, in rat diaphragm and gastrocnemius muscles, 24 h after a single injection of LPS or saline solution. mRNA was measured by PCR. Data represent mean ± standard error of the mean (SEM) for *n* = 7–8 rats/group. * *p* < 0.05 and ** *p* < 0.01 versus their respective control rats treated with saline, # *p* < 0.05 versus diaphragm of rats treated with LPS. Diaph: diaphragm, Gast: gastrocnemius.

**Figure 3 ijms-23-03641-f003:**
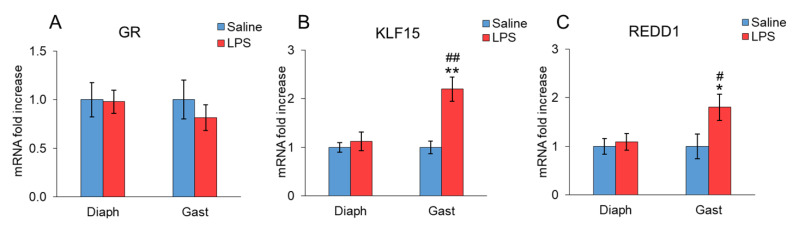
Effect of sepsis induced by LPS injection (10 mg/kg ip.) on (**A**) GR, (**B**) KLF15 and (**C**) REDD1 mRNA, in rat diaphragm and gastrocnemius muscles, 24 h after a single injection of LPS or saline solution. mRNA was measured by PCR. The data represent mean ± standard error of the mean (SEM) for *n* = 7–8 rats/group. * *p* < 0.05 and ** *p* < 0.01 versus their respective control rats treated with saline, # *p* < 0.05 and ## *p* < 0.01 versus diaphragm of rats treated with LPS. Diaph: diaphragm, Gast: gastrocnemius.

**Figure 4 ijms-23-03641-f004:**
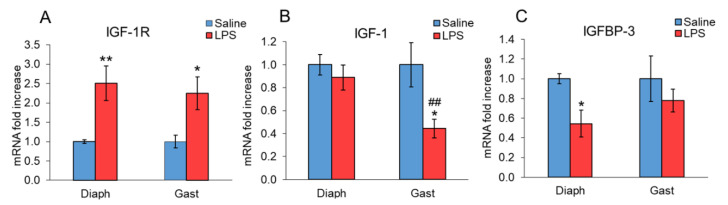
Effect of sepsis induced by LPS injection (10 mg/kg ip.) on (**A**) IGF-1R, (**B**) IGF-1 Ea (IGF-1), and (**C**) IGFBP-3 mRNA, in rat diaphragm and gastrocnemius muscles, 24 h after a single injection of LPS or saline solution. mRNA was measured by PCR. The data represent mean ± standard error of the mean (SEM) for n = 7–8 rats/group. * *p* < 0.05 and ** *p* < 0.01 versus their respective control rats treated with saline, ## *p* < 0.01 versus diaphragm of rats treated with LPS. Diaph: diaphragm, Gast: gastrocnemius.

**Figure 5 ijms-23-03641-f005:**
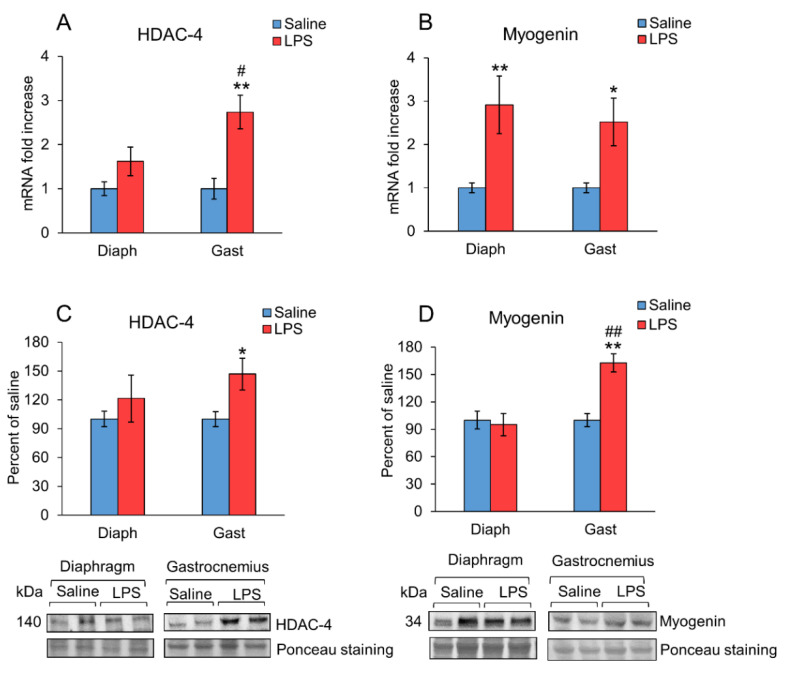
Effect of sepsis induced by LPS injection (10 mg/kg ip.) on (**A**) HDAC-4 and (**B**) Myogenin mRNA, (**C**) HDAC-4, and (**D**) Myogenin in rat diaphragm and gastrocnemius muscles, 24 h after a single injection of LPS or saline solution. mRNA was measured by PCR and proteins by Western blot. Representative Western blots are shown in (**C**,**D** (bottom)). The data represent mean ± standard error of the mean (SEM) for n = 6–8 rats/group. * *p* < 0.05 and ** *p* < 0.01 versus their respective control rats treated with saline, # *p* < 0.05 and ## *p* < 0.01 versus diaphragm of rats treated with LPS. Diaph: diaphragm, Gast: gastrocnemius.

**Table 1 ijms-23-03641-t001:** Effect of lipopolysaccharide (LPS) administration (10 mg/kg ip.) on body weight gain, diaphragm and gastrocnemius weights, serum concentrations of urea, creatinine, glucose, lactate, bicarbonate (HCO_3_^−)^, partial pressure of carbon dioxide oxygen (PaCO_2_), oxygen (PaO_2_), and soluble oxygen SO_2_ and pH in arterial blood. The data represent mean ± standard error of the mean (SEM). ** *p* < 0.01 versus control rats treated with saline.

	Control, *n* = 9	LPS, *n* = 8
Body weight gain (g/24 h.)	−2.0 ± 0.9	−21.9 ± 3.6 **
Diaphragm (mg/100 g b.w.)	206 ± 15	232 ± 17
Gastrocnemius (mg/100 g b.w.)	520 ± 20	517 ± 10
Urea (mg/dL)	22.1 ± 1.7	49.8 ± 5.9 **
Creatinine (mg/dL)	0.208 ± 0.008	0.331 ± 0.03 **
Glucose (mg/dL)	234 ± 8	180 ± 16 **
Lactate (mmol/L)	1.6 ± 0.2	5.6 ± 0.8 **
HCO_3_^-^ (mmol/L)	26.0 ± 0.6	18.0 ± 2.2 **
PaCO_2_ (mmHg)	42.7 ± 3.9	37.8 ± 5.6
PaO_2_ (mmHg)	86.0 ± 6.1	88.0 ± 6.6
SO_2_% (mmol/L)	95.0 ± 1.3	95 ± 0.9
pH	7.39 ± 0.03	7.30 ± 0.04

**Table 2 ijms-23-03641-t002:** Primers for mRNA detection.

Gene	Forward Primer (5′ to 3′)	Reverse Primer (5′ to 3′)
*18 S*	GGTGCATGGCCGTTCTTA	TCGTTCGTTATCGGAATTAACC
*IL10*	AGTGGAGCAGGTGAAGAATGA	TCATGGCCTTGTAGACACCTT
*IL6*	GGAAGTTGGGGTAGGAAGGA	CCTGGAGTTTGTGAAGAACAACT
*TNFα*	TGAACTTCGGGGTGATCG	GGGCTTGTCACTCGAGTTTT
*SOCS3*	CCTCCAGCATCTTTGTCGGAAGAC	CATTCGGGAGTTCCTGGACCAGTA
*MuRF1*	TGTCTGGAGGTCGTTTCCG	AAGTGATCATGGACCGGCAT
*Atrogin-1*	GAACAGCAAAACCAAAACTCAGTA	GCTCCTTAGTACTCCCTTTGTGAA
*IGF-1 Ea*	GCTATGGCTCCAGCATTCG	GGATGAGTGTTGCTTCCGGA
*IGFBP3*	GGAAAGACGACGTGCATTG	GCGTATTTGAGCTCCACGTT
*IGF1R*	GCCTCCAACTTTGTCTTTGC	TCACTGGGCCAGGAATGT
*HDAC4*	CACACCTCTTGGAGGGTACAA	AGCCCATCAGCTGTTTTGTC
*Myogenin*	CCTTGCTCAGCTCCCTCA	TGGGAGTTGCATTCACTGG
*LC3b*	CAGGTTGCCTAGCAGAGGTC	TGTCCTATACACCTGACCTGTTTC
*GR*	AAGAGCAGTGGAAGGACAGC	GCTGGGCAGTTTTTCCTTCG
*KLF15*	TTGTGGGCCAGAAGTTCC	TGCATTTGTGCATTTTGAGAA
*REDD1*	CCAGAGAAGAGGGCCTTGA	CCATCCAGGTATGAGGAGTCTT

## Data Availability

Not applicable.

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
