# Peer review of "Role of Glucocorticoid Signaling and HDAC4 Activation in Diaphragm and Gastrocnemius Proteolytic Activity in Septic Rats"

_ijms, 2022, doi:10.3390/ijms23073641_

Round 1
Reviewer 1 Report
Alvaro Moreno-Ruperez and colleagues examined the effects of sepsis-induced proteolysis on diaphragm and gastrocnemius muscles. The authors’ manuscript was quite simple to show the different expression levels of genes and proteins regarding muscle atrophy between two muscle portions. In addition, the authors tested the pathways between HDAC4 and myogenin. The authors concluded that proteolysis and autophagy were more preferentially enhanced in gastrocnemius muscle than in diaphragm in sepsis model rats, which was caused by lower activation of glucocorticoid signaling and HDAC4-myogenin pathways. However, the manuscript is descriptive. Furthermore, the essence of the manuscript that “proteolysis and autophagy were more preferentially enhanced in gastrocnemius muscle than in diaphragm“ has already been reported by Katherine Bricceno et al. (Human Molecular Genetics 2012, 21, 4448-4459). Most of the experiments were replication studies. The authors should clarify the mechanism underlying the distinct muscle response against sepsis-induced proteolysis.
Sepsis model
The authors described that rat body weight was reduced following LPS-injection. But there was no information on body weight in Table 1.
Some muscle atrophy markers were upregulated in Fig. 1. It would be nice to conduct histochemical analyses. Muscle cross-sections should be stained with antibodies against MuRF-1, LC3b, and atrogin. The sections should be stained with an anti-ubiquitin antibody to show enhancing proteolysis. HE stained sections would also be required to show atrophied myofibers. In addition, immunoblot analysis is required to show the increment of ubiquitinated proteins.
Calpain is also involved in sepsis. The calpain protease activity in the authors’ rat model should be examined.
Figure 4
IGF-1 has splicing variants. Which type of IGF-1 did the authors detect?
IGF-1 is mainly produced in the liver. Did the authors measure IGF-1 concentration in sera?
Muscle fiber type
Slow muscle fibers are more abundant in Diaphragm than gastrocnemius. Did the authors observe muscle fiber type-specific protein degradation in sepsis-induced muscles?
Author Response
The authors described that rat body weight was reduced following LPS-injection. But there was no information on body weight in Table 1.
We agree with the reviewer, but it was an error in this sentence, it is missing the word “gain”. Since LPS reduced the body weight gain over the 24 h of the experiment. As it can be seen in the first row of Table 1, the body weight gain in control was -2.0 ± 0.9 g ± sem, vs - 21.9 ± 3.6 in the rats treated with LPS (P<0.01). Accordingly, that error was corrected in the new version of the manuscript, (see in pg 3, line 109 now is “LPS administration decreased body weight gain (P<0.01)”).
Some muscle atrophy markers were upregulated in Fig. 1. It would be nice to conduct histochemical analyses. Muscle cross-sections should be stained with antibodies against MuRF-1, LC3b, and atrogin. The sections should be stained with an anti-ubiquitin antibody to show enhancing proteolysis. HE stained sections would also be required to show atrophied myofibers. In addition, immunoblot analysis is required to show the increment of ubiquitinated proteins.
Thank you for your suggestion, it would have been very interesting the immunohistochemistry studies, but unfortunately, muscle samples were not collected in the necessary form to perform a histological analysis. In addition, the aim of our study was to elucidate the differences in the mechanisms responsible for ubiquitin-proteasome and autophagy activation between diaphragm and gastrocnemius, but not to study the atrophy in detail. In fact, the study of the effects of sepsis in the myofibers atrophy, comparing diaphragm and a limb muscle (in this case tibialis anterior, TA), has been previously described using different time-periods (doi: 10.14814/phy2.14248). Authors show that the sepsis-induced atrophy of the TA muscle was more prolonged and sustained than the one of diaphragm, demonstrating that this last muscle it is relatively resistant to atrophy. However, these authors do not explore the reason of this difference. These results are part of the basis of our hypothesis and the mentioned reference is described and cited in the introduction (see page 2, lines 46-59).
Calpain is also involved in sepsis. The calpain protease activity in the authors’ rat model should be examined.
As the reviewer pointed out, muscle contractile proteins are degraded by the ubiquitin-proteasome system, autophagy, and proteases such as calpains and caspases. The ubiquitin proteasome system is the predominant protein-degrading system in muscle that is activated during diverse muscle atrophy conditions such as inflammation. The specificity of ubiquitin-proteasome-mediated protein degradation is assured by E3 ubiquitin ligases, such as atrogin-1 and MuRF1, which target structural and contractile proteins (Haberecht-Müller et al. 2021, Biomolecules, 11,1327), for that reason both atrogin-1 and MuRF1 expression are selected as marker of the activity of this system. Autophagy-lysosomal pathway enables the breakdown of proteins to produce amino acids by skeletal muscle that can be utilized in other tissues during catabolic periods, such as starvation (Masiero et al. 2010, Autophagy 6:307–309). In contrast, the overall relevance to increased muscle proteolysis of the calpains and caspases activity in muscle atrophy is not yet clear (Webster 2020, Front Physiol. doi: 10.3389/fphys.2020.597675). In this sense, in critically ill patients with sepsis, the activity of both the ubiquitin-proteasome and the autophagy-lysosomal systems was increased, whereas calpain and caspase activities were not changed (Klaude et al. 2012, Clin Sci (Lond), 122:133-142).
Figure 4
-IGF-1 has splicing variants. Which type of IGF-1 did the authors detect?
It was the IGF-1 Ea, and it is now indicated in table 2 as well as in Fig. 4 legend.
IGF-1 is mainly produced in the liver. Did the authors measure IGF-1 concentration in sera?
We did not measured serum concentrations of IGF-1 as this hormone does not explain the different effect of sepsis on respiratory and limb muscles, since all cell of the body received the same concentration of hormones. Differences in the hormone action must reside in the cellular response, i.e. the receptors or their signaling pathways. Nevertheless, both liver and serum IGF-1 concentration are very sensitive to LPS, as both were significantly reduced at low LPS dosage of 0.01 mg/kg, as we have previously reported (Priego et al. 03, doi: 10.1677/joe.0.1790107).
Muscle fiber type
Slow muscle fibers are more abundant in Diaphragm than gastrocnemius. Did the authors observe muscle fiber type-specific protein degradation in sepsis-induced muscles?
We did not study the specific fiber type degradation, since it was not the aim our study, although we think that it is a very interesting matter. Nevertheless, other authors have addressed this issue. In this way Moarbes et al. (doi: 10.14814/phy2.14248) observed that sepsis elicits time-dependent and muscle- and protein-specific effects on myofibrillar protein levels. These authors observed in diaphragm that MHCIIa and MHCIIx mRNA levels decreased after 24 h of sepsis, MHCIIb levels decreased after 24 and 48 h, whereas MHCI levels remained unchanged. In contrast, in tibialis anterior, of all four MHC isoforms mRNA decreased after 24 h of sepsis. We have added a comment in this in the discussion section (see page 8, lines 230-232).
Reviewer 2 Report
In this paper the authors try to elucidate the mechanism responsible by which muscle atrophy due to sepsis mainly takes place in locomotor muscles rather than in respiratory ones. Sepsis was induced in rats by LPS administration.
Main results: LPS increased TNFα and IL-10 expression in diaphragm and gastrocnemius muscles, whereas IL-6 and SOCS3 mRNA increased only in diaphragm. However, diaphragm showed a lower increase in proteolytic marker expression (atrogin-1 and LC3b) and in LC3b protein lipidation after LPS administration. Moreover, increased the expression of glucocorticoid induced factors, KLF15 and REDD1, and decreased that of IGF-1 in gastrocnemius but not in diaphragm.
They conclude lower activation of both glucocorticoid signaling and HDAC4-myogenin pathways by sepsis can be one of the causes of lower sepsis-induced proteolysis in diaphragm compared to gastrocnemius.
The paper is well written, is concise and shows results with potential interest.
Minor consideration:
Can you better explain the concept "protein lipidation", and its etiopathogenic interest? This expression is used several times in the text and should be defined
Author Response
Can you better explain the concept "protein lipidation", and its etiopathogenic interest? This expression is used several times in the text and should be defined.
We agree with the reviewer, and in the introduction of the manuscript we have explained the role of LC3b in autophagy and the significance of measuring the LC3b-II/LC3b-I ratio (see words in red, page 2, lines 77-83). Microtubule-associated protein light chain 3b (LC3b) is the only known mammalian protein that is specifically associated with autophagosome (Zhang et al. 2017, Curr Protoc Toxicol. 69: 20.12.1–20.12.26). Upon the autophagy induction, the cytosolic LC3b (LC3b-I) is conjugated to phosphatidylethanolamine (PE) to form lipidated LC3b (LC3-II). LC3b-II binds to the expanding isolation membrane and remains bound to complete autophagosome. Therefore, LC3b-II (the lipidated form) is widely used as a marker for autophagy (Mizushima et al., 2010 Autophagy, 3:542–545).

Reviewer 3 Report
The authors investigated about the mechanism responsible of atrophy in muscle proteolysis in sepsis model in rat. The manuscript is almost well written. Overall the topic could be interesting but many details are not clear.
I recommend that the paper be accepted with minor revision:
a) The authors should better describe the rational behind the study.
b) In the introduction section, little previous evidence is provided about the importance of sepsis in several pathologies. Incorporating comparisons with other studies would increase the strength of the paper. Please refer to doi: 10.3390/ijms22111388; 10.7554/eLife.49920.
c) The authors should better emphasize the conclusions.
d) There are some minor grammar issues that should be fixed in order to aid the accessibility of the results to the reader.
Author Response
a) The authors should better describe the rational behind the study.
We thank the reviewer this comment, we have added a paragraph in the introduction describing the rationale behind the study (see page 3 lines 104-107).
b) In the introduction section, little previous evidence is provided about the importance of sepsis in several pathologies. Incorporating comparisons with other studies would increase the strength of the paper. Please refer to doi: 10.3390/ijms22111388; 10.7554/eLife.49920.
In the introduction, we have added the importance of different pathologies induced by sepsis, especially myopathies, as suggested by the reviewer (see page 1 lines 9-12 and page 2, lines 41-45). In fact, we have used one of the recommended references (the second one). The first doi referred must be misspelled, since we were unable to find the reference. We thank the reviewer for the comment as we consider important to highlight the sepsis-associated alterations to strength the article.
c) The authors should better emphasize the conclusions.
As suggested by the reviewer we have added some sentences to emphasize the conclusions (see page 10 lines 333-336).
d) There are some minor grammar issues that should be fixed in order to aid the accessibility of the results to the reader.
A native English speaker has made some correction in the results section.

Reviewer 4 Report
- Please, result and discussion part should be described in detail, to increase reader’s understanding.
- You explained that an increase in HDAC4 expression was found in the skeletal muscle of aged mice, but is there any type of HDAC that increases in the respiratory muscle?
- Looking at Table 1, there seems to be no significant change in gastrocnemius, but can't you confirm muscle wasting even if you measure the longer time after LPS injection?
- In Figure 2C, the expression of IL-6 in gastrocnemius was quadrupled compared to control. Why did you say the difference was not significant?
- Why do you think myogenin's mRNA expression has increased very much in diaphragm, but protein levels have not increased? And wouldn't there be more factors that differ in protein levels from other mRNA expressions as in this case?

Author Response
- Please, result and discussion part should be described in detail, to increase reader’s understanding.
As suggested by the reviewer we have lengthened some parts of the result and discussion sections in order to improve the reader understanding.
2. You explained that an increase in HDAC4 expression was found in the skeletal muscle of aged mice, but is there any type of HDAC that increases in the respiratory muscle?
As discussed in the article, the HDAC4 expression has not been studied so far in any model of LPS-induced muscle atrophy. And, although the importance of HDAC-4 in atrophy has been observed in the skeletal muscle of the limbs, until now it has not been measured in the diaphragm. Other isoforms can be found in the diaphragm (as HDAC1 and 2) but, to our knowledge, their response to sepsis in this muscle has not been studied so far. However, in patients with chronic obstructive pulmonary disease an increase of HDAC4 protein levels was described in diaphragm (Puig-Vilanova et al 2014, PLoS One. 2014; 9(11) doi:10.1371/journal.pone.0111514). We have added a comment about this in the discussion section (see page 9, lines 308-310)
3. Looking at Table 1, there seems to be no significant change in gastrocnemius, but can't you confirm muscle wasting even if you measure the longer time after LPS injection?
Discrepancies between the results of gastrocnemius weights and the atrogenes and autophagic responses, can be explained by the fact that the biochemical events in the skeletal muscle cells occurs earlier before the muscle atrophy is evident, as we have previously reported (Martin et al 2012, J Physiol Pharmacol 63: 649-59). In that study we administered two LPS injections in a lower dosage (1mg/kg, instead of 10 mg/kg), in two consecutive days. As it can be seen in the above figure 2B (see attached file), there were no significant differences in gastrocnemius weight, 4 h after the second and 24 h after the first LPS injection. However, 24 h after de second, (and 48 h after LPS first LPS injection), gastrocnemius weight was significant decreased. The decrease in gastrocnemius weight was also observed 72 and 96 h after de LPS injections, when the body weight gain was normalized in the rats as it is shown in Fig. A. At these times, body weight gain in the rats injected with LPS was similar to that of control group, while gastrocnemius weight continue decreasing at 24 and 72 h, as you can see in Fig. 2 of that manuscript that is shown in the next page.
However two injections of LPS only increased MuRF1 expression in the first 4 hours after LPS injection, whereas the increase in atrogin-1 was observed until 24 hours after LPS administration (see attached file).
4. In Figure 2C, the expression of IL-6 in gastrocnemius was quadrupled compared to control. Why did you say the difference was not significant?
We agree with the reviewer in his observation. The LPS-induced increase in IL-6 mRNA levels of the gastrocnemius was evident. However, differences did not reach statistical significance (P=0.06) probably due to the large standard deviation of this parameter. We have added a comment on this in the results sections (see page 4, lines 140-142).
5. Why do you think myogenin's mRNA expression has increased very much in diaphragm, but protein levels have not increased? And wouldn't there be more factors that differ in protein levels from other mRNA expressions as in this case?
In our experience both mRNA and protein levels normally are modified in a parallel mode. However in the case of myogenin, disparity between myogenin mRNA and protein levels has been previously reported, and this disparity seems to be due to a mechanisms of postranscriptional regulation (Shiraishi et al. 2007. J. Biol. Chem. 282; 9017–9028; Srikuea et al 2010 Clin Exp Pharmacol Physiol. 37:1078-86.).
We have added a comment on this matter in the result section (see page 6 lines 193-194).

Round 2
Reviewer 1 Report
In addition, immunoblot analysis is required to show the increment of ubiquitinated proteins.
The authors did not answer the above query. They should conduct an immunoblot study to show the upregulation of ubiquitinated proteins in muscle tissues in the septic rat.
Calpain is also involved in sepsis. The calpain protease activity in the authors’ rat model should be examined.
As the reviewer pointed out, muscle contractile proteins are degraded by the ubiquitin-proteasome system, autophagy, and proteases such as calpains and caspases. The ubiquitin proteasome system is the predominant protein-degrading system in muscle that is activated during diverse muscle atrophy conditions such as inflammation. The specificity of ubiquitin-proteasome-mediated protein degradation is assured by E3 ubiquitin ligases, such as atrogin-1 and MuRF1, which target structural and contractile proteins (Haberecht-Müller et al. 2021, Biomolecules, 11,1327), for that reason both atrogin-1 and MuRF1 expression are selected as marker of the activity of this system. Autophagy-lysosomal pathway enables the breakdown of proteins to produce amino acids by skeletal muscle that can be utilized in other tissues during catabolic periods, such as starvation (Masiero et al. 2010, Autophagy 6:307–309). In contrast, the overall relevance to increased muscle proteolysis of the calpains and caspases activity in muscle atrophy is not yet clear (Webster 2020, Front Physiol. doi: 10.3389/fphys.2020.597675). In this sense, in critically ill patients with sepsis, the activity of both the ubiquitin-proteasome and the autophagy-lysosomal systems was increased, whereas calpain and caspase activities were not changed (Klaude et al. 2012, Clin Sci (Lond), 122:133-142).
Although the authors insist that calpain is not involved in sepsis, there are many works of literature to show the upregulation of calpain activity in sepsis (PMID: 34884807, PMID: 32413000, PMID: 31944887. Etc). The authors should not ignore these data.
Author Response
In addition, immunoblot analysis is required to show the increment of ubiquitinated proteins. The authors did not answer the above query. They should conduct an immunoblot study to show the upregulation of ubiquitinated proteins in muscle tissues in the septic rat.
As recommended by the reviewer we have performed an immunoblot study of the ubiquitinated proteins. Results are shown in the figure of the attached document.
As it can be seen in the figure, the ubiquitinated protein levels were similar in the rats injected with LPS or saline in both skeletal muscle diaphragm and gastrocnemius. These data are in accordance to those reported by other authors that were also unable to find a significant increase in the ubiquitinated protein levels in skeletal muscle after LPS administration. In this sense, 8 hrs after an injection of LPS (7.5 mg/kg), did not modify polyubiquitination in the gastrocnemius, although there was a trend toward increased polyubiquitination in liver of mice (DOI: 10.1186/1471-2474-15-166). Similarly, Mackenzie et al. observed no modification of ubiquitination in rat muscle 24 hrs after administration of 400 µg/kg LPS (DOI:10.1152/ajpendo.00050.2005). We found an article in Chinese (with an English abstract), that reported an increased in ubiquitin and ubiquitinated protein in the EDL muscle at 2 h and 6 h after endotoxin injection. However, no significant changes were noted in the expression of ubiquitin and ubiquitinated protein at 12 and 24 h. The LPS dosage was not included in the abstract (Chai et al. 2001, Zhonghua Wai Ke Za Zhi. 2001 Sep;39(9):721-3). Therefore, we cannot exclude the possibility that during the first hours after LPS injection it would be possible to detect an increase in ubiquitinated proteins in muscle.
In other models of muscle atrophy, such as denervation (DOI:10.1371/journal.pone.0160839), cisplatin administration (DOI: org/10.1016/j.jnutbio.2022.108953) or age-induced sarcopenia (DOI: 10.1007/s11010-015-2608-7), an increase in ubiquination has been described. Differences between these and our study may be due to the model used, since they are models of long-term atrophy.
As the results obtained do not add relevant information to the paper, we have not included them, but we have added a comment about this matter in the result (see pg. 3, lines 128-130 in red).
Although the authors insist that calpain is not involved in sepsis, there are many works of literature to show the upregulation of calpain activity in sepsis (PMID: 34884807, PMID: 32413000, PMID: 31944887. Etc). The authors should not ignore these data.
In order to analyze the involvement of Calpain 1 in our model, we performed a Western blot of Calpain 1. We also analyzed the ratio of α II spectrin cleaved form (150 KDa) versus the intact form (240kDa) as a marker of calpain activity. As showed in the figure of the document attached no significant changes were found.
Perhaps measuring the calpain activivity as indicated the reviewer (PMID: 34884807) would be more discriminative. We tried to measure calpain activity in both muscles, by a commercial fluorometric assay kit (Calpain Activity Assay Kit, Abcam), but unfortunately we do not have freshly made samples as recommended in the manufacturer’s instructions.
Nevertheless, in the introduction, (pg. 2, lines 56-57 in red), we indicated that the increase in calpain activity is involved in sepsis-induced muscle wasting, as recommended by the reviewer (we also included two of the references mentioned by the reviewer).

Round 3
Reviewer 1 Report
Just let me know information on antibodies the authors used in the reply (company, catalog number).
Author Response
The antibodies used in the determination of ubiquitinated proteins, calpain 1 and alpha 2 spectrin were:
- Ubiquitin antibody (P4G7): from Santa Cruz Biotechnology, Inc (sc-53509). It is a mouse monoclonal antibody against full length denatured ubiquitin. It was incubated with the WB membrane overnight at 4ºC (dilution 1:1000).
- Calpain 1 antibody (A-5): from Santa Cruz Biotechnology, Inc (sc-390677). It is a mouse monoclonal antibody specific for an epitope mapping between amino acids 682-709. It was incubated with the WB membrane overnight at 4ºC (dilution 1:1000).
- Alpha 2 spectrin antibody (C-3): It is from Santa Cruz Biotechnology, Inc (sc-48382). It is a mouse monoclonal antibody against aminoacids 2368-2472 of alpha 2 spectrin. It was incubated with the WB membrane overnight at 4ºC (dilution 1:1000).
In all assays to detect the primary antibody, we incubated the membrane with a secondary antibody (anti-mouse immunoglobulin G, IgG, Amersham Biosciences; Little Chalfont, UK) at 1:2000 during 2 hours at room temperature.
Bands were visualized using peroxidase activity (enhanced chemiluminescent reagent from Amersham Biosciences, Little Chalfont, UK). Band intensities were analyzed by densitometry using Gene Tools Analysis software.